# Biodegradable Magnesium Alloys for Biomedical Implants: Properties, Challenges, and Surface Modifications with a Focus on Orthopedic Fixation Repair

Kevin Koshy Thomas [1], Mah Noor Zafar [2], William G. Pitt [3] and Ghaleb A. Husseini [1,2,4,*]

1   Materials Science and Engineering Program, College of Arts and Sciences, American University of Sharjah, Sharjah P.O. Box 26666, United Arab Emirates; b00085892@alumni.aus.edu
2   Biomedical Engineering Program, American University of Sharjah, Sharjah P.O. Box 26666, United Arab Emirates; g00092920@aus.edu
3   Department of Chemical Engineering, Brigham Young University, Provo, UT 84602, USA; pitt@byu.edu
4   Department of Chemical and Biological Engineering, American University of Sharjah, Sharjah P.O. Box 26666, United Arab Emirates
*   Correspondence: ghusseini@aus.edu

**Abstract:** Biomedical devices made from high-modulus and hardness materials play a critical role in enhancing the quality of life for people with bone-related ailments. While these materials have been successfully used in orthopedic applications, concerns including stress-shielding have necessitated the exploration of alternative solutions. An ideal biomedical implant requires a delicate balance of mechanical performance, corrosion resistance, tissue biocompatibility, and other properties such as tribological performance and osseointegration. This review explores the suitability of biodegradable magnesium (Mg) alloys as a promising material for biomedical implants. It delves into the essential properties of biomedical implants, emphasizing the importance of matching mechanical characteristics with human bone properties to mitigate stress shielding. The corrosion properties of implant materials are discussed, highlighting the need for controlled degradation to ensure the safety and longevity of implants. The focus then shifts to the potential of magnesium alloys as biomedical implants, examining their benefits, limitations, and the challenges associated with their high degradation rates and less-than-satisfactory mechanical properties. Alloying with elements such as aluminum, zinc, and others is explored to improve magnesium alloys' mechanical performance and corrosion resistance. Furthermore, this review discusses surface modification techniques, including chemical conversion coatings and biomimetic deposition, as effective strategies to enhance the corrosion resistance and biocompatibility of magnesium and its alloys. These modifications offer opportunities to improve the long-term performance of magnesium-based biomedical implants. This review provides a comprehensive overview of the properties, challenges, and potential solutions associated with biodegradable magnesium alloys as a promising material for biomedical implants. It underscores the importance of addressing problems related to mechanical performance, corrosion resistance, and biocompatibility to advance the development of safe and effective biomedical implant materials.

**Keywords:** magnesium alloys; biomaterials; biomedical implants

## 1. Introduction

Many biomedical devices are constructed from natural or synthetic materials characterized by high modulus and hardness, often serving as implants to enhance the well-being of people afflicted with damaged or missing bone structures [1,2]. These sturdy materials find potential applications elsewhere in the human body, such as heart valves and intravascular stents, depending on the specific medical needs [3–7]. Elderly people are more likely to experience orthopedic health problems such as arthritis, often necessitating the use of

implanted devices to replace compromised biological structures [8]. Conditions such as osteoporosis, osteoarthritis, and trauma further compound the challenges older patients face, potentially leading to localized pain and physical deterioration in the functionality of hard tissues [9]. Addressing this problem has involved substantial time and effort invested over several decades in the development of various orthopedic implants [10]. Metal-based implants crafted from materials such as titanium-based, cobalt-based, or stainless-steel alloys have emerged as prominent choices in load-bearing orthopedic applications due to their high-modulus mechanical properties. However, research has revealed that these implant materials are susceptible to stress-shielding, a side-effect wherein a high-modulus orthopedic implant bears such a substantial load that the adjacent bone, lacking its typical load, undergoes deterioration [11]. Ideal biomedical implants must possess three fundamental characteristics: mechanical performance, corrosion resistance, and tissue biocompatibility, alongside other essential attributes, including tribological behavior, osseointegration, and non-toxicity [12].

Introducing a biomedical implant into the body involves multiple interactions at the tissue–implant interface. These interactions define the biocompatibility of an implant, significantly impacting its biological and mechanical performance [13–15]. For instance, if an orthopedic material releases toxic elements, it cannot be considered biocompatible. Therefore, selecting materials for biomedical implants necessitates a paramount focus on non-toxic, biocompatible substances that closely mimic the elastic modulus and strength of human bone [16–21].

Apart from materials designed for permanent implants, there is a growing interest in durable biomaterials that can gradually biodegrade over time, allowing for the regeneration of bone or other tissues to fulfill their original functions. Materials such as stainless steel, titanium, and cobalt alloys remain non-degradable throughout the implant's lifespan. In contrast, certain magnesium alloys can be engineered to degrade safely and under controlled conditions within the body. This review examines biodegradable magnesium (Mg) alloys as a promising material for biomedical implants, scrutinizing their bulk and surface properties and their overall performance.

*Key Properties of Biodegradable and Biomedical Implants*

Generally, implants serve as medical devices that facilitate interaction with biological systems [22]. Their proximity to bodily tissues presents both useful medical applications and significant medical implications. As mentioned earlier, an ideal implant material must exhibit requisite mechanical properties, excellent biocompatibility, and a low corrosion rate suitable for the mechanical and biological demands of the application at hand. For example, titanium alloys find use in joint replacements due to their lower elastic stiffness and excellent compatibility with bone [10]. Consequently, the design of a safe and reliable implant material necessitates a meticulous blending of mechanical, chemical, physical, and biological properties to ensure prolonged functionality without the need for replacement surgeries.

A more comprehensive exploration of desirable properties in medical implants and biomaterials, in general, is presented below.

## 2. Mechanical Properties

A critical requirement for any load-bearing implant material is aligning its mechanical characteristics with those of human bone. Whether composed of metal, ceramic, polymer, or their combinations, the material's mechanical performance must be tailored based on tissue characteristics and load requirements at the specific implantation site [23]. To this end, some very general and exemplary mechanical and durability prerequisites are summarized in Tables 1 and 2.

**Table 1.** Examples of mechanical and durability requirements for general biomedical material design [24].

| Mechanical Performance | |
|---|---|
| **Biomaterial** | **Mechanical Characteristics** |
| Hip prosthesis | Strong and rigid |
| Tendon material | Strong and flexible |
| Heart valve leaflet | Flexible and tough |
| Articular cartilage | Soft and elastomeric |

**Table 2.** Examples of durability conditions for biomedical implant design [24].

| Mechanical Performance | |
|---|---|
| **Biomaterial** | **Durability** |
| Bone plate | Six months or longer |
| Hip joint | Ten years or longer under heavy loads |
| Heart valve leaflet | Able to fully flex 60 times/min without undergoing failure for several decades |

The choice of materials for implants is often guided by their specific mechanical properties, tailored to their intended applications within the human body. For instance, Titanium-based alloys are commonly employed in bone fixation devices and high-load scenarios such as hip prosthetics, whereas cobalt-based alloys are preferred for joint replacement [25,26]. An implant material must exhibit an adequate modulus and yield strength to fulfill its intended function. Furthermore, for implants such as hip and knee replacements, robust fatigue resistance under substantial loads is imperative [27]. The key mechanical properties essential for evaluating an implant include tensile modulus, yield strength, hardness, compressive and shear strength, toughness, fatigue strength, and permanent elongation [28]. These mechanical characteristics should closely match those of human bone to withstand decades of use and minimize the phenomenon of stress shielding. Furthermore, implant materials should exhibit outstanding wear resistance, especially for articulating surfaces such as hip and knee prostheses, often associated with surface hardness. Minimizing the formation and propagation of cracks is also vital in high-load applications [29]. Zhang et al. investigated the mechanical properties of biological Mg-Zn-Ca alloy for bone implants before and after high-pressure torsion (HPT) processing. HPT enhances microhardness and tensile strength but reduces toughness, while annealing at 210 °C for 30 min balances microhardness distribution and significantly improves plasticity. The Mg-Zn-Ca alloy exhibited optimal comprehensive mechanical properties after five turns of HPT, followed by annealing [30]. A novel strategy for achieving the optimal balance between the mechanical performance of alloys and meeting load-bearing requirements is by creating porous structures in implant materials. Powder bed fusion techniques, facilitated by lasers, contribute to the generation of low-density magnesium structures, thereby reducing the stress-shielding effect. The advantage of employing this technique over other manufacturing methods lies in the ability to adjust porosity levels compared with alternative additive manufacturing techniques. This technique produces magnesium with controlled porosity, which will exhibit a predetermined external form and internal structure that complements bone rigidity, eliminating the necessity for stress shields during manufacturing. Table 3 shows different mechanical properties of biodegradable and non-biodegradable alloys.

**Table 3.** Mechanical Properties of different Magnesium alloys and non-biodegradable materials [31].

| Material | Young's Modulus (GPa) | Yield Strength (MPa) | Tensile Strength (MPa) | Hardness (HB) |
|---|---|---|---|---|
| Ti-6Al-4V | 55–110 | 760–1103 | 860–965 | 334 |
| Co-Cr | 240 | 500–1500 | 900–1540 | - |
| Stainless Steel | 200 | 170–310 | 540–1000 | 201 |
| Magnesium | 45 | - | 170 | - |
| AZ31 | 45 | 171–176 | 290 | 49 |
| AZ91 | 45 | 160 | 240 | 63 |
| Mg-Ca-Zn | 45 | - | - | 30–60 |

As mentioned above, an elevation in the mechanical properties of the implant, particularly the modulus, in comparison to the surrounding human bone, frequently results in stress reduction within the neighboring bone, which typically would bear the full load [32]. Consequently, regions experiencing reduced loading may suffer from atrophic bone density and mass loss. This bone mass and strength decline can lead to implant loosening or even fracture, necessitating premature revision surgery [33]. To mitigate the impact of stress shielding, innovative implant design approaches that consider material stiffness, geometry, and shape adjustments become imperative [34].

## 3. Corrosion Properties

Since one of the critical properties of Mg alloys is controlled degradation through corrosion, we present a brief overview of corrosion principles. Programmed degradation, often referred to as controlled biodegradation, serves to meet temporary load-bearing requirements without interfering with the healing and regeneration of local tissues. However, once local wound healing has occurred, the implant gradually degrades, slowly transferring the load to the adjacent bone tissue [35]. Conversely, undesired but often normal degradation of any alloy can manifest as surface corrosion, oxidation, and mechanical wear of the implant material. Such undesirable degradation can compromise the structural integrity of the implant, release potentially toxic metallic ions, and generate wear debris particles, resulting in unfavorable biological reactions in neighboring tissues or in mechanical impediment of articulating joints. The preferred model for biodegradable implants within the body involves the degradation or corrosion of metals that are non- to minimally harmful or that can be rapidly eliminated from the body through physiological processes without local concentrations exceeding any toxic thresholds during the corrosion process [36]. A critical consideration when designing biodegradable implants is to achieve a controlled degradation rate that aligns with the timeline of tissue restoration, as depicted in Figure 1. In a controlled degradation scenario, employing biodegradable materials is intended to facilitate the restoration of tissues to their natural state, allowing for transformation and sustained growth. Throughout the degradation process, it is crucial for the implant to maintain its structural integrity until after its mechanical function diminishes. On the pathway to failure, the implant should undergo a controlled and reproducible deterioration without causing undesired side effects. The degradation rate can be regulated by selecting appropriate alloying elements. Incorporating these elements enhances degradation or corrosion resistance up to a certain threshold, beyond which the resistance will decline and controlled failure will progress [37]. Magnesium (Mg) alloys are among the materials employed for biodegradable purposes. This is particularly relevant in hard material prostheses because $Mg^{2+}$ ions are non-toxic at normal concentrations, and the corrosion rate can be controlled [38]. More than half of the magnesium in the human body is already found in the bones. Incorporating magnesium as a significant metallic element in the design of biomaterial alloys can aid in the favorable degradation of a biodegradable implant, as shown in Figure 2, and contribute to bone healing through cellular repair mechanisms [11]. As mentioned, for controlled degradation to be safe, the metal ions released from corrosion need to be non-toxic, but an acceptable alternate is that they are released at a slow enough

rate that the local concentration in the adjacent tissues never exceeds the safe toxic limit. Yet, this may create conflicting objectives since often one of the objectives is to degrade at the same rate as tissue is restored in vivo [39].

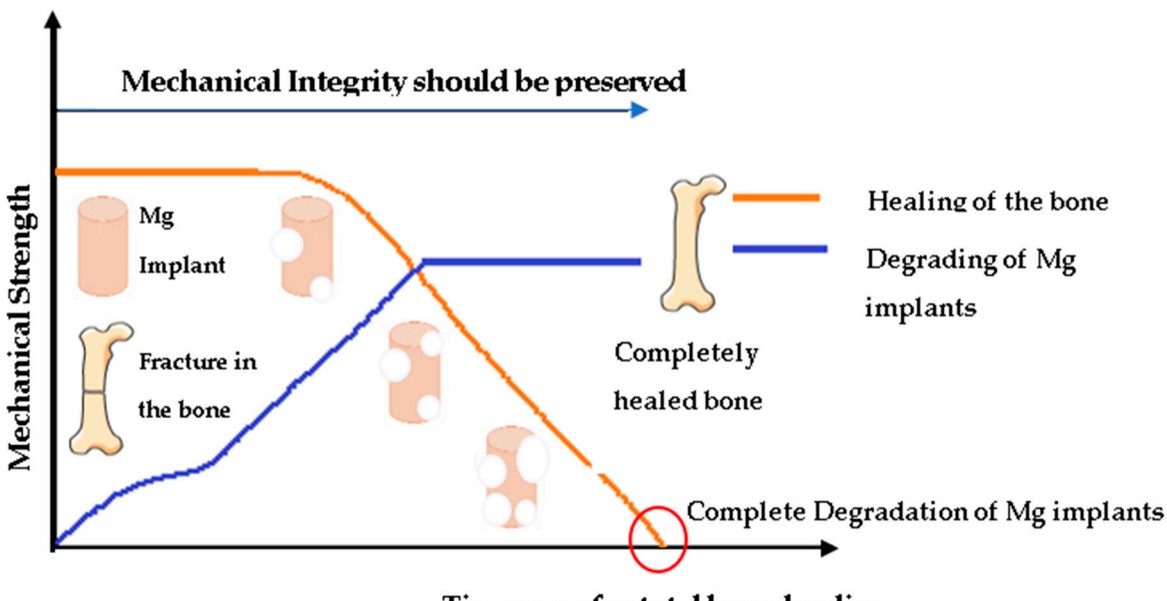

**Figure 1.** Degradation process for biodegradable Mg implant.

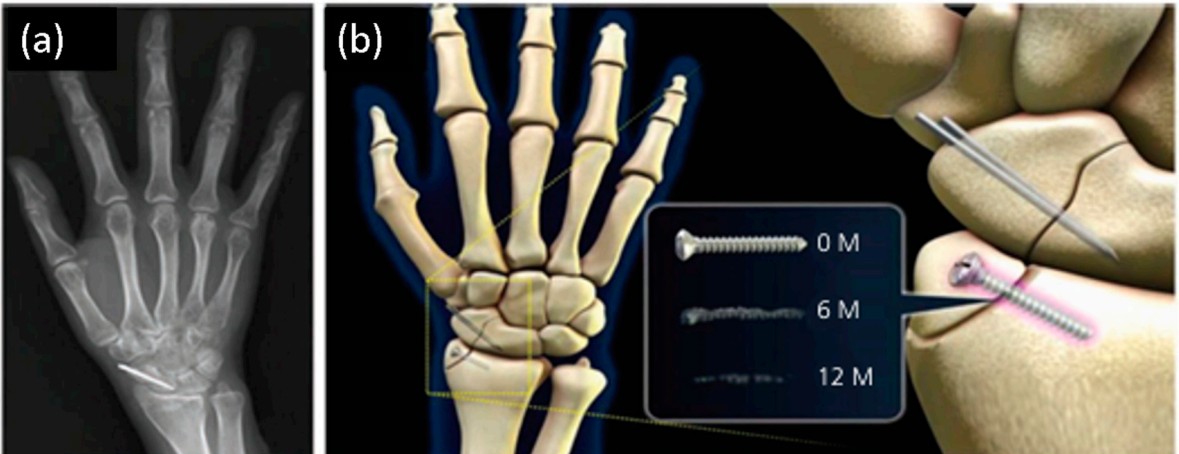

**Figure 2.** A schematic diagram illustrating (**a**) the implantation site and (**b**) magnesium alloy's changes over time at 0.6- and 12-months post-implantation.

The main challenge for Mg and its alloys is to control the rate of degradation, which is caused when it encounters body fluids due to the influence of different factors, such as pH, alloying elements, concentration, and types of ions. This means that magnesium implant materials require sufficient mechanical strength and integrity during their time in the human body. Different studies have revealed that these factors can be controlled by modifying the surface and polymeric coatings. The research literature has shown that there are different types of biopolymers, among which the natural ones stand out due to their biomimetic nature, biocompatibility, and cell proliferation because they provide barriers to limit direct contact between the surface of the Mg and the aqueous biological environment. Medical applications of Mg and Mg alloy coatings are challenging, and achieving a uniform coating and an effective degradation rate is difficult. In many cases, a pretreatment on the Mg surface is required to achieve better adhesion and corrosion resistance. Different surface

modification methods were reviewed to improve the corrosion resistance of magnesium or its alloys. Thus far, many techniques have been developed for the protection of the substrate; however, not all of them are used since some are toxic or repress biocompatibility. The most commonly used technique in natural biopolymers is chemical conversion because it provides excellent adherence, attributed to the formation of chemical bonds with the substrate. A candidate for biomedical implants must satisfy several requirements: it must offer mechanical support, degrade at a reasonable rate to be replaced by the new bone, favor cell adhesion, proliferation, and cellular differentiation, prevent infection, and exert a positive osteogenic effect. The latter makes corrosion prevention for Mg alloys very attractive, with the release of osteoinductive factors and growth factors to speed up the healing process. Figure 3 shows the mechanism for biodegradable implants.

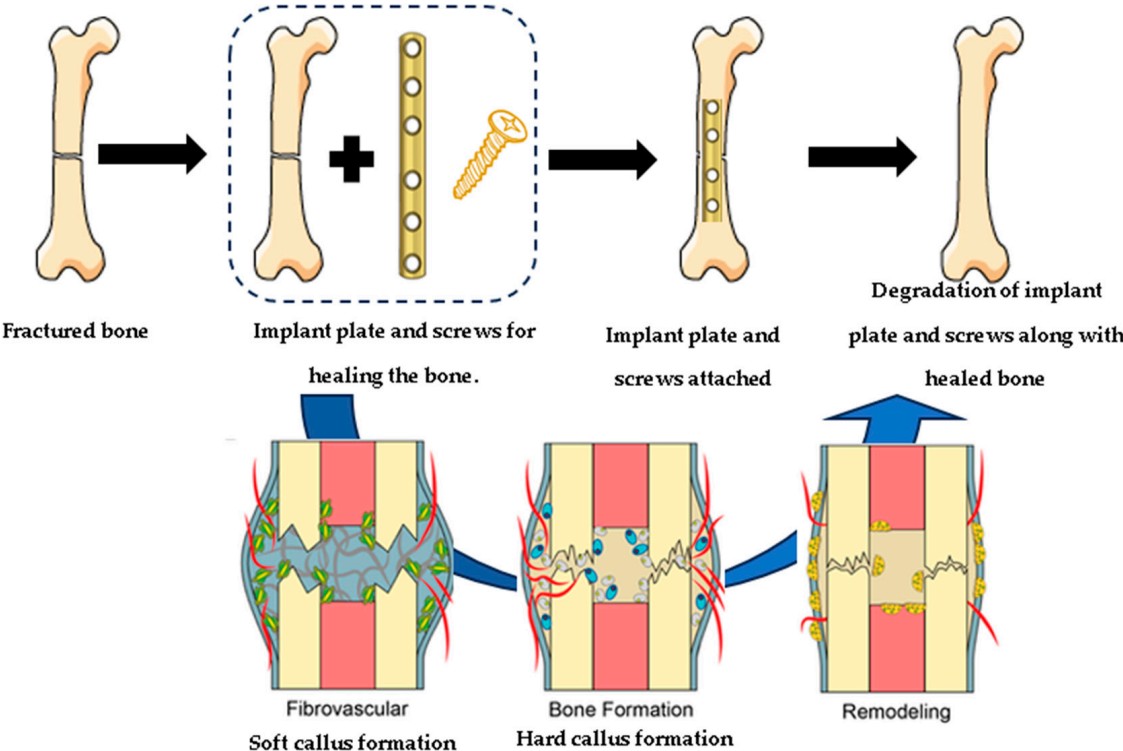

**Figure 3.** The mechanism for biodegradable implants.

Indicators such as pH change, hydrogen evolution, $Mg^{2+}$ release, changes in mechanical strength, or morphological changes are essential in measuring the corrosion behavior of the material. However, mechanical strength changes have not been widely investigated. There are very few investigations of natural coatings on Mg with in vivo assessments, which are necessary for future studies. In order to evaluate the potential of these materials in clinical applications, it is necessary to obtain more substantial evidence of optimal performance, qualitative research, and development, together with the collaboration of clinicians to obtain materials for specific uses. Natural coatings are promising because they protect against corrosion, can be functionalized, improve osseointegration, and the body can metabolize the subproducts of the degradation as drug release. There are many efforts to diminish corrosion, and the progress is promising. However, more studies with other natural polymers, such as fibrin or stearic acid, which are more frequently studied with other metals, are necessary. Furthermore, more studies on drug delivery and the release of osteoinductive factors and growth factors are required.

Undesired degradation of certain implants could be dangerous when placed in the human body if the ions being released are toxic; they could cause discoloration of the implant or the adjacent tissue [40,41]. Furthermore, unwanted corrosion may compromise the mechanical properties of the implant. Types of unwanted corrosive behavior in implants in-

clude galvanic, pitting, crevice, and fatigue corrosion. In one such example from a fractured stainless-steel implant inside a patient's thigh, Farzad et al. found extensive structural damage, including crack initiation from pitting, intergranular surface cracking inside the crevice, and more [42]. Galvanic corrosion has been observed in dental implants when two metal elements (such as in a Co-Cr alloy) are combined to manufacture implants [43]. In orthopedic applications, crevice corrosion has been observed in bone fixation implants at the contact interface between bone plates or between screws and bone tissue. Studies have shown that a larger crevice size leads to a higher implant corrosion rate [44]. Due to the adverse effects of toxic products on patients leading to implant restriction or denial, there is a growing need for corrosion-resistance materials to ensure the longevity of implant materials in physiological conditions. The electrochemical resilience of these alloys must be strong enough to accommodate corrosion rates that can adversely affect the composition of these materials. These alloys should be capable of withstanding low pH values and exhibit minimal degradation under physiological conditions to ensure the mechanical integrity of the implant remains intact until the bone or tissue has adequately healed [37].

Du et al. studied the microstructure, tensile properties, and corrosion behavior of Mg-4Zn-1.2Y-0.8Nd alloy undergoing thermal extrusion processes. The findings indicated that elevating extrusion passes leads to improved corrosion resistance. Moreover, extrusion ratio and pass alterations affect mechanical properties, with varying impacts on tensile strength and elongation.

Clinical observation of a 29-year-old female patient's 1-year follow-up using X-rays revealed a complete degradation with bone healing after 1 year. Khiabani et al. assessed the properties of magnesium-based alloys, specifically those alloyed with aluminum, zinc, calcium, zirconium, yttrium, and rare-earth elements, for biomedical implant applications. These degradable biomaterials offer lightweight composition, fracture toughness, and an elastic modulus close to bone density. They can be alloyed to develop biodegradable metals with controlled corrosion rates; however, the study also emphasizes carefully selecting alloying elements to maintain biocompatibility [37]. Furthermore, applying Mg-pretreated periosteum (M–P) in anterior cruciate ligament (ACL) reconstruction presented an innovative approach to enhance tendon-bone healing.

Different types of Mg implants have varying degradation rates and mechanical properties. The time it takes for the implant to degrade and for the bone to integrate with the surrounding tissue can depend on specific characteristics of the Mg alloy. Moreover, the initial condition of the bone, the nature of the injury, and the patient's overall health can affect the healing process. A too-rapid degradation rate may compromise the mechanical integrity, whereas too-slow degradation may impede the natural healing and remodeling of the bone. From a toxicological standpoint, an implant material should not release toxic ions into the body unless such release is intentionally desired. Thus, the implant material should exhibit controlled chemical stability in the complex chemical environment of the human body [45]. Nevertheless, released toxic elements entering body tissues are virtually inevitable, even if implant materials are believed to possess robust corrosion resistance [46,47]. Studies have shown that the release of metal ions due to implant wear in the human body is a leading cause of toxic effects related to implant materials, including tissue sensitization, damage to local blood vessels supplying the bone, bone necrosis, and more [48]. The deposition of non-matching metallic ions at the implantation site can trigger allergic and toxic responses, leading to inflammatory reactions and tissue damage [49]. Generally, the $Mg^{2+}$ ion is naturally present throughout the body and plays a crucial role in numerous biochemical reactions. Typically, Mg is not a cause for concern; however, pure Mg lacks adequate mechanical properties and thus is frequently alloyed with other elements to enhance strength and modulus while reducing brittleness. Biodegradable alloys are generally considered safe, biocompatible, and non-cytotoxic when the viability of cells adjacent to corroding metals surpasses 70% compared with a control group. For instance, the Mg-Zr-Sr-Sc alloy is considered biocompatible because cell viability remains at least 83% [50]. Toxicity is typically attributed to released ions other than $Mg^{2+}$. The

assessment of biocompatibility is primarily carried out through in vivo experiments. These experiments replicate physiological conditions within laboratory settings and test tubes. In vivo tests encompass two main approaches: direct-contact and indirect-contact experiments. Cytotoxicity assays and cell proliferation tests are commonly employed to evaluate cell behavior [37]. Table 4 shows the corrosion rates for different types of Magnesium alloys with HA coating.

**Table 4.** Corrosion rate of different Magnesium alloys [11].

| Magnesium Alloy with % of Hyaluronic Acid (HA) | Degradation Behavior | Immersion Medium and Maximum Time | Immersion Test Type |
|---|---|---|---|
| Mg-6Zn-5%HA | 15% reduction in weight loss after 30 days of immersion time | SBF, 30 days | Static |
| Mg-3Zn-5%HA | Enhanced degradation rate by 40% for the initial 14 days of immersion | SBF, 56 days | Static |
| Mg-5%HA | Huge decrease in corrosion rate (around 75%) | PBS, 96 h | Static |
| AZ91–20%HA | Improved degradation rate | Artificial Sea Water, 24 and 72 h | Static |

When Mg is inserted into the biological environment, $Mg^{2+}$ ions are produced on the Mg surface due to the anodic reaction of the metal. Simultaneously, $H_2$ and $OH^-$ ions are produced by the cathodic reaction occurring on other nearby surfaces, and the $OH^-$ ions react with the $Mg^{2+}$ to form $Mg(OH)_2$ film. The reaction produces a $Mg(OH)_2$ film covering the magnesium alloy surface. In a biological setting, chloride ions are always present, which react with magnesium hydroxide to form soluble $MgCl_2$, exposing fresh Mg surface for further corrosion [24]. These reactions are summarized below.

$$\text{Anode}: \text{Mg} \rightarrow \text{Mg}^{2+} + 2e^- \tag{1}$$

$$\text{Cathode}: 2H_2O + 2e^- \rightarrow 2OH^- + H_2 \tag{2}$$

$$\text{Product formation}: Mg^{2+} + 2OH^- \rightarrow Mg(OH)_2 \tag{3}$$

## 4. Magnesium Alloys for Biomedical Implants

Achieving successful implant integration requires outstanding biocompatibility; thus, medical implants must meet stringent standards for materials, including being non-toxic, non-carcinogenic, non-pyrogenic, non-allergic, non-inflammatory, and hemocompatible. In the laboratory, it is essential to simulate the physiological conditions when performing in vitro tests, including cytotoxicity assays and cell proliferation studies, to evaluate the potential material effects on the host organism and assess the biocompatibility of the biodegradable implants before implantation [37].

Magnesium (Mg) alloys, which are generally biodegradable, have garnered considerable attention in the biomedical field in recent years. In addition to mechanical characteristics such as human bone, Mg shows excellent biocompatibility. As mentioned above, Mg is a vital supplement for keeping the human body healthy and promoting osteo-growth [2,51,52]. Additionally, during corrosion in the biological environment, $Mg^{2+}$ is released, creating non-toxic magnesium hydroxide, hydroxyl ions, and hydrogen gas (as shown above), which are removed from the body via the kidneys [53–57]. On the other hand, the main disadvantages of using Mg alloys are the rapid degradation rates and decrease in mechanical properties, which may cause implant failure faster than the healing process [58]. Moreover, hydrogen gas emitted from corrosion may create gas pockets that cause the adjacent tissues to separate, and the $OH^-$ ions may cause surface alkalization and potential cell damage [59,60]. Additionally, the mechanical properties of Mg could be

improved by alloying. Thus, there are various challenges to tackle in developing Mg alloys. The advantages and disadvantages of Mg material are summarized in Tables 5 and 6.

**Table 5.** Benefits of Mg alloys in medical implants [24].

| Advantages | Description |
|---|---|
| Reduced density and elastic stiffness | Density and elastic stiffness are similar to human bone. |
| Higher specific strength | Strength to weight ratio is roughly 35–260 KNm/kg. |
| Machinability | Mg has good machining capability to achieve accurate dimensions and processing into complex shapes. |
| Stress shielding effect | The elastic stiffness of Mg is very close to bone. |
| Biocompatibility | Mg is shown to have osteogenic functions. |
| Degradability | Mg naturally degrades in the body, which is favorable to the patients. |

**Table 6.** Limitations of Mg alloys in medical implants [24].

| Disadvantages | Description |
|---|---|
| Low mechanical properties | Implants must be able to endure specific loads and deformation. It is difficult for Mg to meet medical demands in strength and plasticity. |
| High degradation rate | It leads to premature loss of mechanical integrity and implant supports. |
| Hydrogen generation | Hydrogen release creates air bubbles in the surrounding tissues. |

Even though Mg has many benefits as an implant material, pure Mg is not recommended for biomedical applications due to its higher corrosion rate and insufficient mechanical properties. Moreover, Mg shows low ductility because of the lack of slip characteristic in its hexagonal closest packed (hcp) structure. These problems are averted by alloying, which creates microstructural changes and adjusts surface potential between phases, improving mechanical properties and corrosion resistance [61,62]. Common alloying elements for Mg are aluminum (Al) and zinc (Zn), as they contribute to hardness, strength, and castability. Lithium provides low density and high solid solubility to Mg alloys. Moreover, Li can change the formability of Mg alloys by changing the crystal structure of Mg from hcp to the body-centered cubic (BCC) [56]. Alloying elements refine grains and optimize the type, size, and distribution of the second phase, which reduces the corrosion rate of Mg alloys. Furthermore, alloying elements create passive films to impede further corrosion. Presently, aluminum (Al), zinc (Zn), manganese (Mn), calcium (Ca), strontium (Sr), zirconium (Zr), and neodymium (Nd) are commonly used as alloying elements. The influence of these elements on the Mg alloys is shown in Table 7.

Magnesium alloys have attracted substantial attention in the biomedical field owing to their impressive attributes, including remarkable strength, low density, and outstanding osteogenic biocompatibility, as highlighted in Table 7. Employing pure Magnesium alloys as implant materials comes with specific disadvantages. First, their mechanical properties fail to meet the necessary standards for implant materials, and their rapid resorption results in mechanical instability prior to complete bone healing. Second, magnesium alloys exhibit accelerated degradation rates, particularly in chloride-rich environments such as human bodily fluids. The generation of degradation rates gives rise to problems such as tissue inflammation. As mentioned, a notable challenge arises due to its considerable vulnerability to corrosion, originating from its remarkably low standard electrode potential of $-2.37$ V. Finally, implants made from pure magnesium alloys experience non-uniform corrosion, leading to premature failure. Among the concerns in the domain of biomedical implants, the corrosion behavior of Mg implants plays a preeminent role. When Mg is immersed in an aqueous milieu, a pivotal electrochemical process occurs, as highlighted in Equations (1) through (3), and underscores the significance of managing corrosion dynamics associated

with Mg alloy im-plants for biomedical applications. Tafel plots for Magnesium alloys are shown in Figure 4 [63].

**Table 7.** Effect of alloying elements on Mg alloy performance [24].

| Alloying Elements | Biocompatibility | Corrosion Resistance | Mechanical Performance |
|---|---|---|---|
| Al | ■ Al is neurotoxic. It can cause Alzheimer's disease and damage muscle fibers. | ■ It is beneficial in providing corrosion resistance. | ■ Increases strength and plasticity. |
| Zn | ■ Non-cytotoxic and good biocompatibility. | ■ Corrosion resistance decreases with higher Zn content. | ■ Zn participates in solid solution strengthening, increasing strength with increasing Zn content. |
| Mn | ■ Cytotoxic and neurotoxic. | ■ Provide good corrosion resistance. | ■ It gives higher yield strength and reduces tensile strength and ductility. |
| Ca | ■ An essential component of human bone. | ■ Corrosion resistance decreases with increasing Ca content. | ■ With increasing Ca content, strength increases, and plasticity decreases. |
| Sr | ■ Sr is a vital component of human bone. It also aids in bone formation. | ■ Corrosion resistance of Mg alloy drops with increasing Sr content. | ■ The strength of the alloy increases with increasing Sr content. |
| Zr | ■ Good biocompatibility and bone-bonding ability. | ■ Corrosion resistance decreases with increasing Zr content. | ■ Grains undergo refinement, increasing strength and plasticity. |
| Nd | ■ Cytotoxic at high concentrations but has good biosafety at low concentrations. | ■ Improves corrosion resistance. | ■ Forms new phases, grain refinement, and improvement in mechanical performance. |

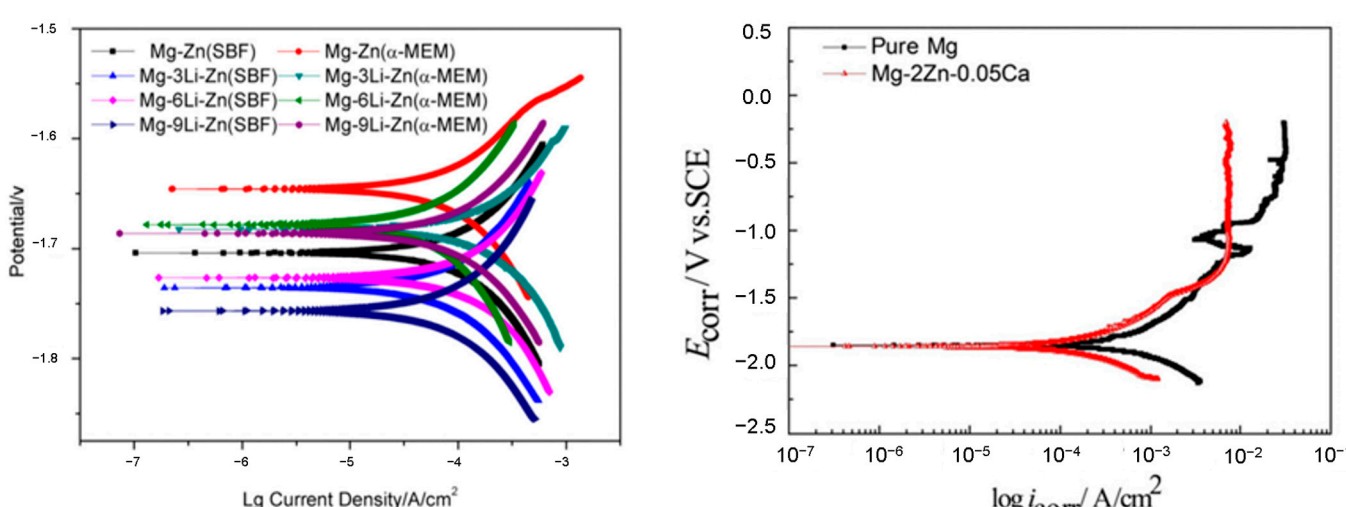

**Figure 4.** Tafel plots for Magnesium alloys [64–66].

Because of magnesium's inherent low strength, alloying becomes the predominant approach for enhancing its mechanical properties. However, introducing alloying elements gives rise to secondary phases that may expedite corrosion. Moreover, the release of alloy ions may play distinct roles in bone healing. Apart from their supporting function, the released ions from magnesium alloys during biodegradation exhibit additional effects, such as antibacterial and antitumor properties. In contrast to non-biodegradable materials, magnesium alloys stand out as a suitable choice for biodegradable implants due to their biocompatibility and the absence of toxicological tissue responses. These magnesium alloys present lower probabilities of releasing cytotoxic ions, thereby preventing stress shielding

and local tissue inflammation [67,68]. Adding aluminum (Al) in commercial magnesium (Mg) alloys is the most prominent strategy, contributing to increasing the overall corrosion resistance of Mg alloys. AE21, AZ32, and AZ91 are some examples of Al-enriched Mg alloys with remarkable mechanical and corrosion resistance properties that are used for biomedical applications. However, Al's potential in the development of Alzheimer's disease raises concerns and underscores the requirement of intricate equilibrium in biomedical applications. Zinc (Zn), on the other hand, is vital for the human body and enhances the material strength of Mg-Al alloys. Mg-Zn alloys are highly biocompatible. Moreover, it forms protective layers that effectively slow down degradation. Mn is an essential trace element for the human body with the potential to form a protective Mn-containing oxide film that inhibits Cl-ion infiltration, rendering it a commonly employed element in Mg alloys for biomedical applications. Mg-Ca alloys favor bone implantation, reducing potential differences and improving corrosion resistance. Furthermore, Strontium (Sr) also shares chemical properties with Mg and Ca; it is found in bones and promotes bone formation. Other elements found in Mg alloys for biomedical applications include Zirconium (Zr), Silicon (Si), and Lithium (Li), which foster bone bonding and corrosion resistance and add ductility to vascular stents in biomedical applications, respectively.

To summarize this section, alloying elements play a crucial role in controlling the degradation rate of magnesium alloys. The composition and concentration can be tailored to achieve the desired balance between sufficient strength for implant functionality and a degradation rate that matches the tissue healing process. The degradation rates of Mg alloys vary and determine their corrosion characteristics. Aluminum and zinc, when alloyed with magnesium, increase the oxidation rate, whereas rare earth elements decrease the oxidation rate of magnesium alloys. Combining aluminum and zinc in specific ratios yields an alloy with an overall improved performance, balanced corrosion resistance, mechanical strength, and biocompatibility [69–71].

## 5. Surface Modification Used for Mg Alloys

In addition to tweaking the bulk alloy chemistry to avoid toxicity and manipulate mechanical properties, the toxicity can be reduced by forming coatings on the surface, which impedes the release of toxic ions so that the local ion concentration can be kept below toxic levels and the corrosion rate is also inhibited. When MgO is exposed to ionizing radiation, energy is transferred to the MgO crystal, promoting the ejection of electrons from their bound states, which interact with impurity atoms in the crystal lattice. The ionization-radiation-induced alterations in MgO extend beyond impurity states, encompassing the creation of defects such as the V1 center, which influences materials' electronic and optical properties. However, MgO is radiation resistant and relatively stable compared with other materials. The radiation-induced damage to MgO is minimal, with zero compromise to its functionality. This is due to its ability to emit light when exposed to ionizing radiation and has been employed as a scintillator material in radiation detectors. However, the response of MgO to pulsed laser irradiation revealed defect formation, surface fracture, and rapid vaporization [72,73]. On the other hand, electronic irradiation-induced electronic defects within the $MgF_2$ crystal affect the material's electronic structure and properties [74].

Coatings allow the use of alloys with excellent mechanical properties, even if they contain some toxic ions. Such surface modifications may also enhance the bioactivity of these biodegradable materials [62]. The precise impact of different alloying elements incorporated into the Mg substrate on coating kinetics and the resultant properties affecting corrosion resistance and biocompatibility remains somewhat elusive. Nevertheless, it is acknowledged that the coating's hardness directly influences biocompatibility. Several surface modification techniques, such as surface coating, chemical conversion coatings, biomimetic deposition, micro-arc oxidation coating, sol-gel coating, and ion implantation, have demonstrated enhancements in prolonging the in vivo lifespan of Mg implants. Yet, utilizing coatings and surface modifications in a controlled environment faces several considerations. These encompass challenges such as the constraints associated with rapid

corrosion, the generation of substantial hydrogen gas volumes, and the potential accumulation of hydrogen bubbles in gas pockets. Addressing these "surface" factors is imperative before contemplating the large-scale production of Mg alloys for use as implants [75]. Improving the mechanical and tribological properties of magnesium alloys, such as tensile strength, hardness, corrosion resistance, and wear resistance, can be accomplished by generating a surface composite layer with reinforcement particles through friction stir processing (FSP). Despite numerous attempts to develop multi-functional coatings for magnesium-based alloys, addressing the issue of localized corrosion without compromising mechanical strength over an extended period remains an unresolved challenge. The search for an optimal solution continues [76,77]. Two general types of surface modification are used for Mg alloys: (a) surface coating preparation and (b) surface microstructure modification.

A.    Surface coating preparation:

The oxide film formed on the Mg alloy surface is generally weak, which cannot chemically or mechanically protect the alloy for long exposures. Hence, there is a need to make a protective film on the surface utilizing chemical, physical, mechanical, and biological or biomimetic techniques [78].

● Chemical conversion coatings:

These coatings are formed by electrochemical or chemical reactions of Mg-based alloys within an electrochemical bath to form a layer that often contains fluoride, phosphates, carbonate, or chromate groups [68,79,80]. An insoluble compound film with reasonable adhesion formed on the Mg surface can protect the Mg alloy from light mechanical stress and harsh aqueous environments; these initial layers can improve the adhesion of further coatings. These methods are easy to execute and are often used in biomedical applications. Fluoride and phosphate coatings are used for biomedical Mg alloy surfaces [65,81,82]. Normally, fluoride coatings are produced in hydrofluoric acid (HF) by reaction with Mg alloys [63,83]. The main component of fluorine coating is magnesium fluoride ($MgF_2$), which is water-insoluble and spontaneously deposits on the Mg surface. $MgF_2$ films increase corrosion resistance and improve cellular response and biocompatibility [64,72]. For phosphate coatings, zinc phosphate and calcium phosphate are used for Mg alloys because of water insolubility, high-temperature resistance, corrosion resistance, and excellent biocompatibility [73,84–86]. Meng et al. [87] conducted a comparative investigation into the corrosion behavior and biocompatibility of various chemical conversion coatings applied to magnesium alloy surfaces. The results indicated that the NaOH coating exhibited the highest levels of hydrophilicity and corrosion resistance. Additionally, the surface treated with NaOH demonstrated the most favorable behavior for endothelial cell growth.

● Biomimetic deposition

This process simulates physiological apatite mineralization in nature and deposits bio-ceramic coatings on the substrate surface, as shown in Figure 5. The benefits of this technique are (a) ease of adjusting coating composition, phase, and crystallinity, (b) capability of coating on porous or complex-shaped implants, and (c) having a simplified method of incorporating biologically active agents or drugs into apatite coatings through coprecipitation. Hence, the biomimetic method is widely used for metallic biomaterials [88,89]. Biomimetic deposition can effectively enhance corrosion resistance and biocompatibility. Dong et al. give an example of forming a biomimetic calcium phosphate film on a Mg alloy [90,91]. The biomimetic deposition of CaP on the poly(dopamine) film, followed by graphene oxide coating, demonstrates a multi-step process for creating a sophisticated material with tailored mechanical, electrical, and biocompatible properties. Pan et al. [92] applied a biomimetic Ca-P coating onto ZK60 magnesium alloys and observed a significant improvement in corrosion resistance when tested in simulated body fluid.

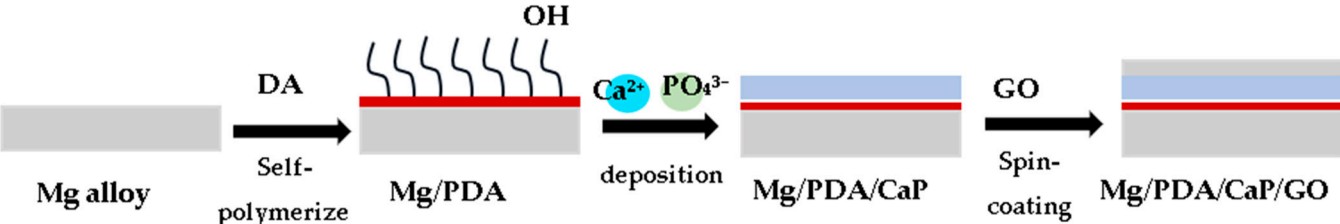

**Figure 5.** Biomimetic deposition of CaP on the surface of a poly(dopamine) (PDA) film in addition to subsequent graphene oxide coating [91].

- Micro-arc oxidation coating

Micro-arc oxidation (MAO), otherwise known as plasma electrolytic oxidation (PEO), is a high-voltage plasma-assisted anodic oxidation process developed from traditional anodizing to create ceramic-like coatings [93,94]. The details of MAO are summarized in Figure 6. MAO has been explored in a variety of fields due to its high efficiency, increased bonding strength between coating and substrate, and minimal restrictions on the surface shape of the workpiece [95–98]. The main limitation of MAO is its inability to provide long-term surface protection [94]. For biocompatibility and biological activity, MAO exhibits higher bonding strength with the substrate due to its dense interior; this porous outer layer is useful in protein adsorption, osteoblast adhesion, and bone tissue regeneration, which is often desirable for biological applications [24].

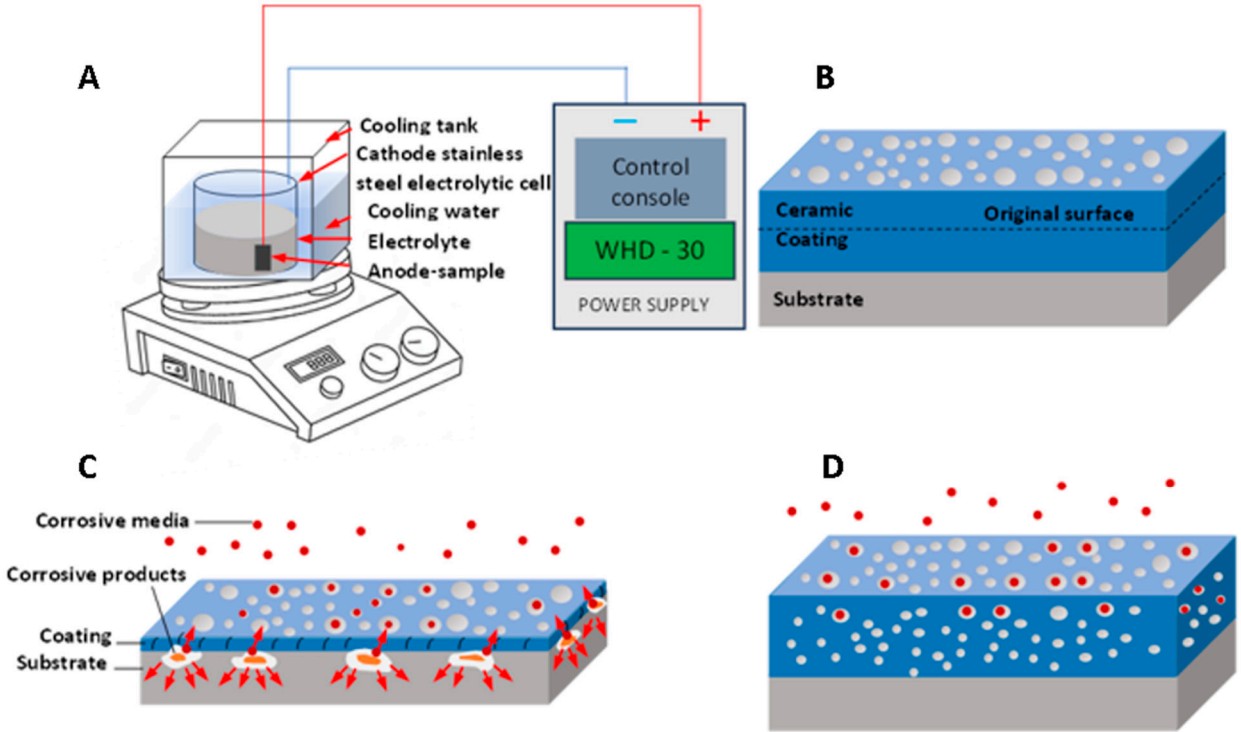

**Figure 6.** (**A**) MAO device, (**B**) growth model of MAO coating, (**C**) thin coating with through pores, (**D**) thicker coating with complex pores.

Yang et al. [99] regulated AZ31B alloy's corrosion and degradation behavior using MAO coatings with varying zinc phosphate concentrations. The AZ31B alloy coated with MAO coatings exhibited improved corrosion resistance in simulated physiological conditions. The addition of zinc phosphate optimized the degradation rate, reduced $Mg^{2+}$ and $OH^-$ release, and alleviated corrosion product formation. After a 56-day immersion, weight loss decreased from 24.37% to 5.22%. Zinc phosphate conversion into $Zn(OH)_2$

restrained infiltration corrosion, fostering a favorable microenvironment for cell activities. The zinc phosphate-doped MAO coating also enhanced wear resistance, making it a promising choice for Mg-based bone fixation implants.

- Sol-gel coating

The sol-gel process, also known as chemical solution deposition, has been used widely in material science and ceramic engineering. This technique initially uses a chemical solution as a precursor to produce an integrated network of discrete particles or network polymers [100,101]. Generally, the sol-gel process involves four steps: (1) hydrolysis, (2) condensation and polymerization of monomers for the formation of chains and particles, (3) growth of particles, and (4) accumulation of polymer structures followed by a continuous network formation in a liquid medium which increases viscosity for gel formation [102,103]. The schematics of preparation for sol-gel coatings are shown in Figure 7. The technique has the benefits of low cost, low processing temperature, and the ability to coat different materials with complex shapes [99,104].

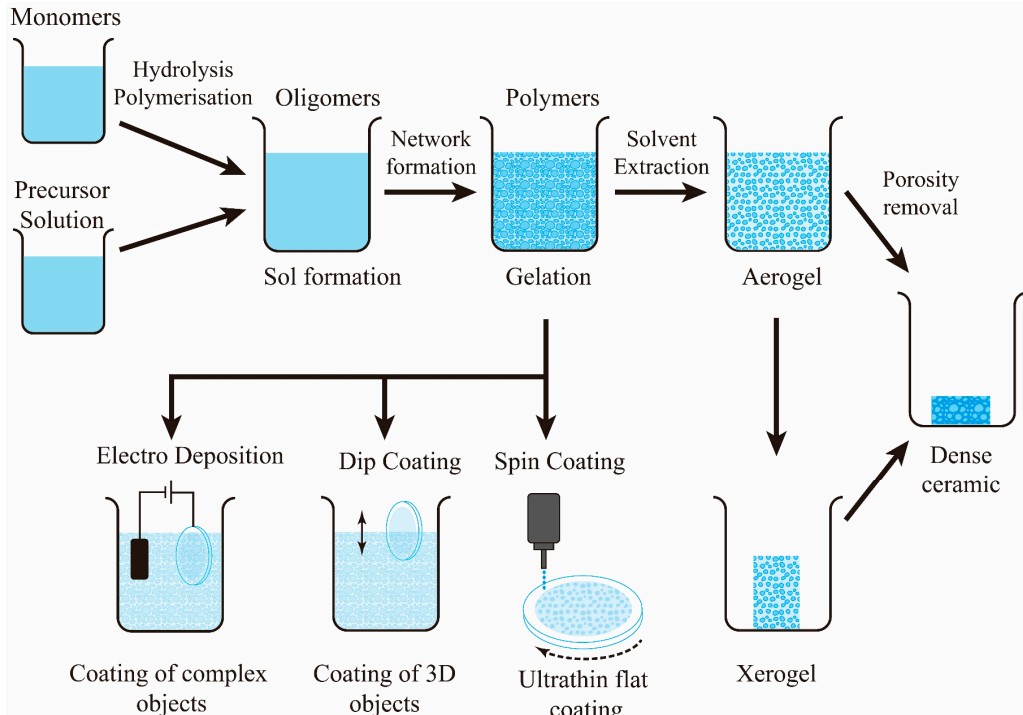

**Figure 7.** Sol-gel coating process. For Mg alloy implants, the object would be subjected to dip coating or electro-deposition.

While adding bioactive silica-glasses (58S and 68S) on AZ91D alloy through the sol-gel technique, Omar et al. [105] discovered that, after 17 days of immersion, 58S provided superior corrosion protection. Based on cell attachment and proliferation tests, AZ91D alloy with either 58S or 68S coatings is deemed suitable for use as temporary implant material.

- Ion implantation:

Ion implantation is a surface modification technique in which target materials are converted to an ion beam in a vacuum, which is later sputtered onto a modified material. Finally, a layer with a specific composition and structure is formed on the surface of the substrate [99,106]. Implanting specific ions onto the Mg substrate can increase the corrosion rate and improve mechanical performance and biocompatibility. For the biodegradable material (Mg alloy), metal ions from iron, cerium, zinc, zirconium, and strontium, and non-metallic ions, for instance, carbon, oxygen, and nitrogen are currently used for ion implantation [86,89,91,102,107–110]. The advantages of this technique include

convenience, controllability, and flexibility. In addition to improving physical and chemical properties, ion implantation improves alloys' biological and antibacterial abilities [24]. Wu et al. [111] used titanium ion implantation to change the surface of pure magnesium and create a self-layered hydrothermal film. Results showed that the inner layer of the film provided good corrosion protection, while the outer layer improved the cell adhesion and cytocompatibility properties.

B.    Surface microstructural modification:

This technique involves the deformation of the metal surface through mechanical processing so that the surface acquires a microstructure and mechanical performance different from the bulk matrix material. Mechanical processing increases the mechanical characteristics, surface hardness, and corrosion resistance of Mg alloys by processes that lead to grain refinement and changes in the distribution of second phases or intermetallic compounds [88,94]. The process usually does not involve chemical reactions. Below are the details of microstructural modification techniques used for Mg alloys.

- Surface mechanical grinding treatment:

Surface mechanical grinding treatment, also known as surface mechanical attrition treatment (SMAT), is a surface nano-crystallization technique that refines grains in the nanoscale and forms gradient nanostructures without changing material composition. It has a noteworthy effect of improving the corrosion resistance of Mg alloys. After the SMAT process, the microstructure of Mg alloys is fine and uniform, the surface is comparatively smoother, and the corrosion rate is considerably lower [97,112]. The main problem of SMAT is reduced degradation resistance due to the crystal defect density caused by attrition balls [98,101,113,114]. So, SMAT is less effective in improving the biomedical Mg alloy performance [24].

- Shot Peening:

Shot peening is a surface modification technique that introduces compressive residual stress on the Mg surface, similar to the SMAT technique. The shot peening process creates a plastically deformed zone with an extended and refined grain structure [101,115]. Mhaede et al. [116] discovered that microhardness and degradation resistance could be improved through shot peening by refining grains and increasing coating density. Peral et al. [117] proved that higher surface roughness developed from shot peening favors rapid degradation. Similar to SMAT, shot peening is limited in improving the biological function of Mg alloys. Several other strategies have recently been researched.

- Laser surface modification

This technique is a productive method for modifying a material's surface by melting the surface by scanning it with a high-intensity laser beam (refer to Figure 8). The advantages are high treatment efficiency, zero pollution, and low material consumption [103,104,118,119]. Currently, laser surface melting, laser cladding, and laser surface alloying are extensively used in surface engineering [106,108,120–122]. For the Mg alloys, laser surface melting is currently used to improve the alloys' mechanical performance and corrosion resistance [109,123]. In addition, laser modification increases the cytocompatibility and corrosion resistance of Mg alloys by modifying the substrate's surface microstructure [24].

- Friction stir processing

Friction stir processing (FSP) is a technique similar to friction stir welding, developed by Mishra [124], and is used for mechanical property improvement and surface composite fabrication of light alloys such as Mg alloys [110,125–127]. FSP uses a non-deformable tool that is forcibly inserted into the workpiece and revolved in a stirring motion as the device is pushed laterally to the alloy. FSP is a technique that can produce refined grains and uniform microstructures and enhance mechanical performances in base materials. FSP is also used to improve the corrosion resistance of Mg alloys by optimizing surface

microstructure or creating a composite layer on the metal surface in biomedical applications [88,112–115,128–130]. FSP creates severe plastic deformation, making grains refined and eliminating surface defects. Moreover, combining FSP with other technologies or reinforcement materials enhances mechanical and biological properties [24].

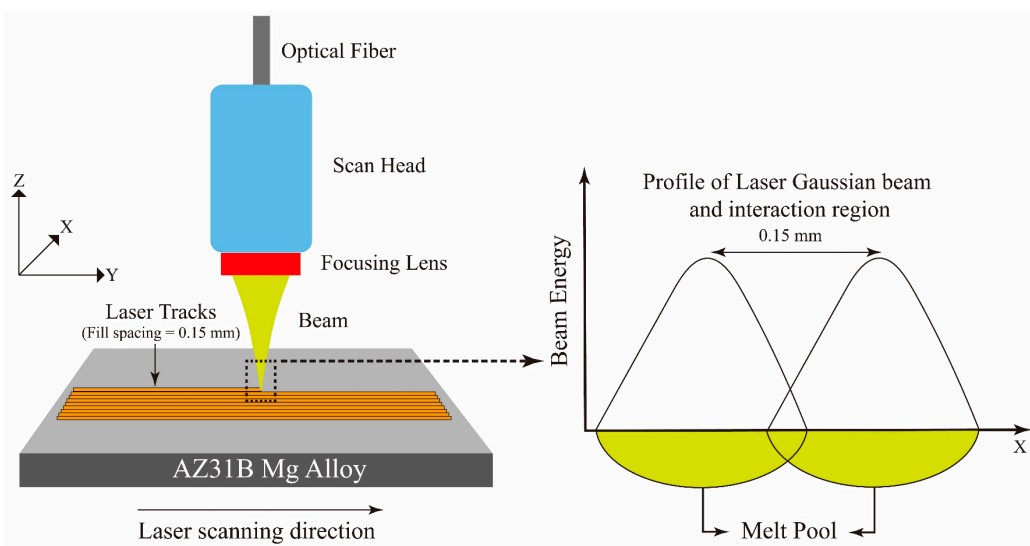

**Figure 8.** Laser surface modification process.

To summarize, Table 8 highlights the types of coatings used on magnesium alloys and their benefits.

**Table 8.** Summary of types of coatings used for biodegradable magnesium alloys and their benefits.

| | | |
|---|---|---|
| Surface Coating | Chemical conversion coatings | • Initial layers improve the adhesion of coatings in the next pass<br>• Easy execution<br>• Improved corrosion resistance and biocompatibility |
| | Biomimetic Deposition | • Ease of adjusting coating composition, phase, and crystallinity<br>• Capable of coating on porous or complex-shaped implants<br>• Enhance corrosion resistance and biocompatibility |
| | Micro-arc oxidation coating | • Higher efficiency<br>• Increased bonding strength between coating and substrate<br>• Minimal restriction on the shape of the implant<br>• Improved biocompatibility |
| | Sol-gel coating | • Lower cost<br>• Low processing temperature<br>• Can coat different materials with complex shapes |
| | Ion implantation | • Controllability and flexibility<br>• Improves the biological and antibacterial properties of implant alloys |

**Table 8.** *Cont.*

| Surface microstructural coating | Laser surface modification | • High treatment efficiency<br>• Zero pollution<br>• Low material consumption<br>• Increases cytocompatibility and corrosion resistance by modifying the microstructure of the alloy |
| --- | --- | --- |
| | Friction Stir Processing | • Improve the mechanical and biological properties of the Mg alloy<br>• Improve corrosion resistance of Mg alloys |
| | Surface mechanical grinding treatment | • Improves corrosion resistance of magnesium alloys |
| | Shot peening | • Improves corrosion resistance of magnesium alloys |

## 6. Summary

Understanding the complex interactions at the tissue–implant interface is crucial for assessing biocompatibility. Implants that release toxic elements are deemed unsuitable for medical use, underlining the importance of focusing on non-toxic, biocompatible materials with mechanical properties akin to those of human bone. While traditional metal-based implants have shown promise, they often produce stress-shielding, which can lead to bone deterioration and necessitate revision surgery. Innovative implant designs that consider material stiffness, geometry, and shape are required to mitigate these challenges.

The emergence of biodegradable materials, such as Mg alloys, presents an exciting avenue for metallic biomedical implants. These materials offer the advantage of controlled degradation, which allows them to temporarily support load-bearing requirements while facilitating the healing and regeneration of local tissues. Moreover, Mg alloys can corrode in a manner that is not harmful, and the resulting byproducts are naturally eliminated from the body. Nonetheless, Mg alloys face challenges related to their rapid degradation rates and concomitant loss of mechanical properties over time. Alloying with elements such as aluminum (Al), zinc (Zn), manganese (Mn), calcium (Ca), strontium (Sr), zirconium (Zr), and neodymium (Nd) can enhance both mechanical performance and corrosion resistance, making them suitable candidates for biomedical applications. Surface modification techniques, including chemical conversion coatings and biomimetic deposition, further improve the corrosion resistance and biocompatibility of Mg alloys. These techniques play a pivotal role in enhancing the longevity and effectiveness of Mg-based implants.

**Author Contributions:** Original draft writing and visualization: K.K.T. and M.N.Z. Reviewing and editing: W.G.P. and G.A.H. All authors have read and agreed to the published version of the manuscript.

**Funding:** This research was funded by the Dana Gas Endowed Chair for Chemical Engineering, American University of Sharjah Faculty Research Grants (FRG20-L-E48, FRG22-C-E08), Sheikh Hamdan Award for Medical Sciences MRG/18/2020, and Friends of Cancer Patients (FoCP).

**Data Availability Statement:** Not applicable.

**Acknowledgments:** The authors would like to acknowledge the financial support of the American University of Sharjah Faculty Research Grants, the Al-Jalila Foundation (AJF 2015555), the Al Qasimi Foundation, the Patient's Friends Committee-Sharjah, the Biosciences and Bioengineering Research Institute (BBRI18-CEN-11), GCC Co-Fund Program (IRF17-003) the Takamul program (POC-00028-18), the Technology Innovation Pioneer (TIP) Healthcare Awards, Sheikh Hamdan Award for Medical Sciences (MRG/18/2020, Friends of Cancer Patients (FoCP), and the Dana Gas Endowed Chair for

Chemical Engineering. The work in this study was supported, in part, by the Open Access Program from the American University of Sharjah and does not represent the position or opinions of the American University of Sharjah.

**Conflicts of Interest:** The authors declare no conflict of interest.

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
