# Peer review of "Biodegradable Magnesium Alloys for Biomedical Implants: Properties, Challenges, and Surface Modifications with a Focus on Orthopedic Fixation Repair"

_applsci, doi:10.3390/app14010010_

Round 1

Reviewer 1 Report

Comments and Suggestions for Authors

Although this is a rather interesting and important review article that can be recommended for publication, however some comments/questions should be taken in to account.

1.     The review contains fairly qualitative descriptions of the problems discussed, but contains almost no actual quantitative information. In this regard, the review cannot serve as a desktop reference that would be useful in the daily routine work of interested researchers.  It would be extremely useful if the authors tried to correct this deficiency by adding some Tables with actual data.

2.     An important aspect that was not considered in the review is the resistance (of Mg compounds) to radiation exposure. Without the latter, it is no longer possible to imagine any medical examination.  In this regard, from solid state radiation physics, we know that the behavior of the Mg compounds here is different. For example, MgO is stable with respect to X-ray and gamma irradiation, but magnesium fluoride MgF2 is no longer stable and degrades over time. On the other hand, the interaction of laser radiation raises the problem of surface integrity.

Chen, Y., & Sibley, W. A. (1967). Study of ionization-induced radiation damage in MgO. Physical Review154(3), 842.

Lisitsyn, V. M., Lisitsyna, L. A., Popov, A. I., Kotomin, E. A., Abuova, F. U., Akilbekov, A., & Maier, J. (2016). Stabilization of primary mobile radiation defects in MgF2 crystals. Nuclear Instruments and Methods in Physics Research Section B: Beam Interactions with Materials and Atoms374, 24-28.

Webb, R. L., Jensen, L. C., Langford, S. C., & Dickinson, J. T. (1993). Interactions of wide bandgap single crystals with 248 nm excimer laser radiation. I. MgO. Journal of applied physics74(4), 2323-2337.

Reviewer 2 Report

Comments and Suggestions for Authors

The review is comprehensive, interesting, and well-structured. I recommend the publication in this form.

Reviewer 3 Report

Comments and Suggestions for Authors

This paper studies Biodegradable Magnesium Alloys for Biomedical Implants: Properties, Challenges, and Surface Modifications.

The English is good.

1. In the part "The key mechanical properties essential for evaluating an implant include tensile
modulus, yield strength, hardness, compressive and shear strength, toughness, fatigue
strength, and elongation [28]. " (rows 103-104), you have 28 sites, before 25, 26. Please check carefully and modify.

2. In Figure 2, on Ox, if you have time, please put the time span for total bone healing.

3. In  the text: "Even though Mg has many benefits as an implant material, pure Mg is not rec-
ommended for biomedical applications due to its higher corrosion rate and insuffi-
cient mechanical properties. Moreover, Mg shows low ductility because of the lack
of slip characteristic in its hexagonal closest packed (HCP) structure. These problems
are averted by alloying, which creates microstructural changes and adjust surface
potential between phases, improving mechanical properties and corrosion resistance
[70], [71]. Common alloying elements for Mg are aluminum (Al) and zinc (Zn), as
they contribute to hardness, strength, and castability [68]. Lithium provides low den-
sity and high solid solubility to Mg alloys. Moreover, Li can change the formability
of Mg alloys by changing the crystal structure of Mg from HCP to the body-centered
cube (BCC) [69]." , is the same thing as point 1.
Please check carefully and modify.

4. Your cite 99 is missing from the text; please check carefully and modify it.

I recommend a major revision.

Comments on the Quality of English Language

Need English spelling check.

Reviewer 4 Report

Comments and Suggestions for Authors

Dear Authors, 

The manuscript submitted for review focusses on presenting the most relevant issues related to magnesium-based implants. Material issues, especially the achievement of appropriate mechanical and corrosion properties, are essential to consider a material for use in an implant at all. The theme of the manuscript is very well suited to its content. The authors have done an extensive literature review in terms of gathering information on magnesium alloys. The manuscript is written clearly and logically. It also reads very well from the language side. I believe that the content of the manuscript could be supplemented/enriched with more information on the Al-Mg combination, especially in terms of the effect of aluminium ions on the increased risk of dementia diseases. At present, one gets the impression that magnesium-aluminium alloys are almost advertised in the manuscript as the best solution.

From the point of view of the solubility of salts containing chloride ions, equation (4) should be simplified to an equation that resembles the dissociation of magnesium hydroxide. Hence, in principle, one can consider omitting this equation and only describing it in detail in the manuscript.

In Table 5, to avoid the problem of when and whether to use the dot mark at all, you can use the bullet pointing of individual sentences with hyphens or other characters.

The words/terms in Figure 5 and Figure 6 are blurred. Of course, one can guess what they mean, nevertheless I think they should be enlarged. 

The notation for the hexagonal packed structure uses lowercase letters (hcp).

What does the statement 'low local ion concentration' mean specifically? What numerical values are used to talk about this? Such statements are overly general. They do not allow reference to numbers. Do the authors have specific data describing the concentration values of ions released from magnesium alloys? If so, it would be worthwhile enriching the text of the manuscript with such data. Line 235.

From my point of view, I think you can replace the word that describes the patients as 'individuals' with 'people'. In this way, we present the topic not in terms of a disease case but of an individual person's story.

I think you might consider replacing a few words with other words: 'encounter'>'experience', 'issues'>'problems', 'previously mentioned'>'mentioned above', 'dependable'>'reliable', 'achieving'>'to achieve', 'demonstrated'>'shown', 'no'>'zero', Lines: 47, 47, 81, 118, 85, 151, 163, 366.

Sincerely yours, 

Reviewer

Reviewer 5 Report

Comments and Suggestions for Authors

I have thoroughly reviewed the manuscript titled "Biodegradable Magnesium Alloys for Biomedical Implants: Properties, Challenges, and Surface Modifications" and find it to be a comprehensive exploration of a topic with significant implications for the field of biomedical materials. The work is well-structured and addresses key aspects of biodegradable magnesium alloys for implants. I appreciate the depth of analysis and the clarity of the presentation. However, to enhance the manuscript further, I would like to raise some advanced and scientific questions for your consideration:

(1) The manuscript touches upon the importance of matching mechanical characteristics with human bone properties to mitigate stress shielding. Could you delve deeper into the specific strategies or considerations for achieving an optimal balance between the mechanical performance of magnesium alloys and the load-bearing requirements of orthopedic applications?

(2) The controlled degradation of magnesium alloys is emphasized as a critical factor for ensuring the safety and longevity of implants. Can you provide more insights into the mechanisms governing the controlled degradation, and how these mechanisms might be influenced by the alloying elements such as aluminum, zinc, and others?

(3) The manuscript rightly highlights the significance of biocompatibility in biomedical implants. Could you elaborate on the methods and standards used for assessing the biocompatibility of biodegradable magnesium alloys? Additionally, how does the degradation process of these alloys impact the surrounding tissues at the molecular and cellular levels?

(4) The review briefly mentions the importance of tribological performance in biomedical implants. Could you provide more details on how the surface modifications, such as chemical conversion coatings and biomimetic deposition, influence the tribological behavior of magnesium alloys? Are there specific challenges or opportunities in this regard?

(5) The surface modifications discussed in the manuscript offer opportunities to improve the long-term performance of magnesium-based biomedical implants. Can you elaborate on the potential durability of these surface modifications under various physiological conditions and their effectiveness in maintaining the intended properties over an extended period?

(6) In the section discussing the benefits, limitations, and challenges of magnesium alloys, a comparative analysis with other biodegradable materials, as well as non-degradable materials commonly used in implants, could provide valuable context. How does the overall performance of magnesium alloys compare to alternative materials, considering aspects such as mechanical properties, biocompatibility, and degradation rates?

(7) The manuscript emphasizes the biodegradability of magnesium alloys for biomedical implants, highlighting the controlled degradation as a crucial factor. Given the inherently degradable nature of these alloys, could you elaborate on the necessity and specific purposes of developing coatings and implementing surface modifications to reduce corrosion rates?

(8) Moreover, the mention of clinical studies indicating the dissolution of bare magnesium implants in the human body within 17-24 months raises an interesting question. In light of these dissolution rates and the natural biodegradability of magnesium alloys, could you provide insights into the clinical significance and practical implications of applying coatings and surface modifications? How do these modifications extend the lifespan or alter the degradation kinetics of magnesium implants in vivo?

(9) Additionally, it would be valuable to explore the specific challenges or considerations in translating the findings from controlled laboratory settings to real-world clinical applications. Are there factors beyond the controlled environment that may influence the efficacy of coatings and surface modifications in mitigating corrosion in diverse patient populations?

(10) In the spirit of enhancing the depth and completeness of your review, I would like to draw your attention to some notable review papers that, in my opinion, could contribute significantly to the context of your work. I believe incorporating insights from these papers would enrich the overall discussion: 

[DOI: https://doi.org/10.1680/jemmr.18.00048]

[DOI: https://doi.org/10.1680/jsuin.20.00057]

[DOI: https://doi.org/10.3390/jmmp6060158]

Integrating insights from these sources could provide a more comprehensive understanding of the current state of research in the field of biodegradable magnesium alloys for biomedical implants.

Comments on the Quality of English Language

Overall, the manuscript is well-written with clear and concise language. However, I recommend thorough proofreading for minor grammatical and syntax adjustments to enhance the fluency and coherence of certain sentences. Additionally, please ensure consistency in terminology and use of scientific language throughout the manuscript. A final round of English language editing will further refine the clarity and professional presentation of your valuable research.

Reviewer 6 Report

Comments and Suggestions for Authors

In the present paper, the authors try to review the properties, challenges and surface modification of Mg alloys for biomedical implant. Though they have exhibited some recent progress on related topic, there are still some questions.

1. The authors titled the paper as “Biodegradable Magnesium Alloys for Biomedical Implants: Properties, Challenges, and Surface Modifications”. Based on the title, it seems that they try to review the Mg alloy used in all biomedical fields, such as vascular intervention therapy, orthopedic repair etc. However, they mainly focus on the orthopedic fixation repair. Such a content is far from the title. They are suggested to revised the title or content.

2. In the section of introduction, the authors briefly introduce the requirement of properties for biomedical materials. Though it could reflect some information, however it deviates from the characteristics of the biodegradable Mg alloy. The authors should focus on the feature of the Mg alloy and introduce its possible application fields and related requirements. The present descriptions on hip joint, heart valve leaflet are not appropriate.

3. The Figure 1 is not suitable here, because the Mg alloy could not act as the permanent implant. Such a schematic is meaningless.

4. In the section of corrosion properties, the authors are suggested put more attention on the features of Mg alloys. Due to the low strength of Mg, the alloying is the primary method to improve the mechanical properties. However, the alloying elements would produce the secondary phases which accelerates the corrosion. In addition, the releasing of alloy ions could act different role during born healing. The authors are suggested to refer the recent researches “Effect of extrusion process on the mechanical and in vitro degradation performance of a biomedical Mg-Zn-Y-Nd alloy. Bioactive Materials 2020,5 (2): 219-227” “Magnesium-pretreated periosteum for promoting bone-tendon healing after anterior cruciate ligament reconstruction. Biomaterials 2021,268: 120576”.

5. In the section of surface modification, the authors are suggested to make clear their aim. Moreover, there is some methods are not the coating, they should not emphasize the coating technology. In addition, they should compare the effects of different surface modification methods by table or figure.

6. In the section of surface modification, the authors are suggested to some results of the recent researches while not schematics. They could refer to the recent researches “Optimization of the in vitro biodegradability, cytocompatibility, and wear resistance of the AZ31B alloy by micro-arc oxidation coatings doped with zinc phosphate. Journal of Materials Science & Technology (In press) https://doi.org/10.1016/j.jmst.2023.09.019  and “NaF assisted preparation and the improved corrosion resistance of high content ZnO doped plasma electrolytic oxidation coating on AZ31B alloy. Journal of Magnesium and Alloys (In press) https://doi.org/10.1016/j.jma.2023.02.008.     

7. There should be some prospects on the biomedical Mg alloys implant based on the authors review, which could improve the manuscript greatly.

8. The present summary is some tedious and could be refined and concise. 

Reviewer 7 Report

Comments and Suggestions for Authors

The work by Thomas et al. (Biodegradable Magnesium Alloys for Biomedical Implants: Properties, Challenges, and Surface Modifications) reviews the biodegradable magnesium (Mg) alloys as a promising material for biomedical implants. The authors evaluated the properties, challenges, and potential solutions associated with biodegradable magnesium alloys as a promising material for biomedical implants.

In general, the manuscript is well-organised and easy to follow. I think the report provides valuable information to readers. I recommend the publication of the report after some points given below are addressed.

Please explain the content of Figure 4.

Please provide some representative studies with relevant figures to keep readers’ attention to figure out the usage of the material. 

Comments on the Quality of English Language

Minor editing of English language required.

Round 2

Reviewer 1 Report

Comments and Suggestions for Authors

The authors have successfully improved the original version of their manuscript, responding constructively to all the comments/recommendations of the reviewer.  Therefore, the article can be recommended for publication.

Author Response

Thank you for your comment. 

Reviewer 3 Report

Comments and Suggestions for Authors

For the Review: "Biodegradable Magnesium Alloys for Biomedical Implants: Properties, Challenges, and Surface Modifications with a focus on orthopaedic fixation repair.", I recommend accepting the present form.

Comments on the Quality of English Language

The quality of English is good, just minor adjustments.

Author Response

Thank you for your comment.

Reviewer 5 Report

Comments and Suggestions for Authors

The authors have adequately addressed the concerns raised by this reviewer. The revised version is deemed acceptable in its current state.

However, a minor issue related to the references requires attention. Reference numbers #38 and #48 have been duplicated. In the text, you referred to reference #48 as "Khiabani et al.," to maintain reference order, I propose substituting the following reference for reference #38: "Ghazanfari H, Hasanizadeh S, Eskandarinezhad S, Hassani S, Sheibani M, Torkamani AD, Fakić B. Recent progress in materials used towards corrosion protection of Mg and its alloys. Journal of Composites and Compounds. 2020; 2(5):205-14."

Comments on the Quality of English Language

English can be double-checked for clarity.

Reviewer 6 Report

Comments and Suggestions for Authors

The authors have revised the manuscript. Though it has been improved, some parts still need to be revised further.

1. In the content, the biodegradable or corrosion mechanism could be be give for better understanding the shortcoming of Mg based alloy, which could be put in section 3.

2. In the section 3, the authors summarize the effect of alloying elements. Actually, except the function of supporting, the released ions during biodegradation also have more effect, such as antibacteria, antitumor, etc. If the authors mainly focus on the orthopedic fixation repair, the related functions could be considered, because the review is really some concise. They could refer the recent researches Corrosion and in vitro cytocompatibility investigation on the designed Mg-Zn-Ag metallic glasses for biomedical application. Journal of Magnesium and Alloys 2023 (In Press) https://doi.org/10.1016/j.jma.2022.09.025 .

3. In the content, the effect of biodegraded Mg alloy on cells or tissue could be discussed. Some results could be displayed in the content, which improves the manuscript further.

4. In the coating, the typical results could given for better understanding. The schematics are not enough to support the effect of the corresponding improvement in corrosion behavior.

5. In the references, there are some errors. For example, the reference 49 is same as the reference 72. The reference 97could be updated for its volume and page number.  

6. The authors are suggested to check the manuscript thoroughly for avoiding the errors.

Author Response

Thank you for your comment. Please find the attached file.

Round 3

Reviewer 6 Report

Comments and Suggestions for Authors

The authors have revised the manuscript partly. It is improve and could be accepted.

Author Response

Thank you for your thorough examination of our manuscript.